

# Research on intelligent forecasts of flight actions based on the implemented bi-LSTM

Xin Hua[1] and Xuejie Yang[2]

[1] Aviation University of Air Force, Changchun, China
[2] Changchun Guanghua University, Changchun, China

## ABSTRACT

Rapid identification of flight actions by utilizing flight data is more realistic so the quality of flight training can be objectively assessed. The bidirectional long short-term memory (bi-LSTM) algorithm is implemented to forecast the flight actions of aircraft. The dataset containing the flight actions is structured by collecting tagged flight data when real flight training is exercised. However, the dataset needs to be preprocessed and annotated with expert rules. One of the deep learning (DL) methods, called the bi-LSTM algorithm, is implemented to train and test, and the pivotal parameters of the algorithm are optimized. Finally, the constructed model is applied to forecast the flight actions of aircraft. The training's accuracy and loss rates are computed. The duration is kept between 1 through 3 h per session. Thus, the development of training the model is continued until an accuracy rate above 85% is achieved. The word-run inference time is kept under 2 s. Finally, the proposed algorithm's specific characteristics, which are short training time and high recognition accuracy, are achieved when complex rules and large sample sizes exist.

## INTRODUCTION

Rapid forecasting of the flight actions of aircraft based on flight records is a new way of objectively assessing the quality level of flight training operations when data-driven methods are implemented. In the conventional sense, however, three issues exist when flight movements are forecasted. First, the constructed rules are so complex that the personnel in charge of recognizing them need to have a high level of expertise. Otherwise, it is difficult to accurately identify them, resulting in erroneous recognitions, for example, the movements of jack flights from a vast quantity of flight data sets. Second, the parameters implemented for judgment rules for the flight actions of aircraft are not accurate enough to interpret all actions; namely, only a few of them can be interpreted accurately. Third, the interpretation process takes a long time; namely, the speed of interpretations is generally slow. Therefore, it is necessary to construct a model that intelligently forecasts several types of flight actions automatically. Parameters can be estimated as closely as possible to overcome problems faced by human screening and evaluation stages. Since improving pilots' flight techniques is important, the objective is to develop better training modules for pilots's training. Thus, hidden dangers are eliminated and the factors causing accidents can

Corresponding author
Xuejie Yang,
yangxuejie1010@sina.com

be pinpointed. Hence, higher flight safety is achieved, when jack-up flight maneuvers. Finally, weak points and tendencies in the full training programs are better detected.

## RELATED WORK

The identification methods of flight actions are divided into two groups, namely, data-driven and knowledge-based approaches (*Shengli, 2022*). The recognition of flight actions that utilize rules refers to a system composed of domain experts. They are artificial prior knowledge, which is extracted as datasets that contain flight characteristics and knowledge. A series of generated rules is represented. Thus, the system needs to have a database composed of recognition knowledge of flight actions. Then, the original flight data must be compared with the constructed rules by implementing data mining or reasoning methods. Eventually, the better recognition stage of flight actions will be realized (*Zhe, 2022*). For example, the US Navy utilizes the V-22 action standard rules (*Shengli, 2022*) to identify the flight actions of H-60 helicopters and applies the derived results to monitoring the structural fatigue of helicopters (*Barndt, Sarkar & Miller, 2007*). While implemented effectively, it also has significant drawbacks (*Ni et al., 2005*). For instance, the number of rule bases reaches up to 1,200, and the rule design is very complicated to verify outcomes. Thus, it cannot be applied to all cases comprehensively. Common action biases of flight actions lead to erroneous identifications, overlapping flight actions, and cross-over transitions that require further processing efforts (*Jin, Wang & Hu, 2017*).

However, more advanced tools have appeared recently. For example, Artificial Intelligence (AI) technologies have matured to compete with previously implemented methods. The research on action recognition in flight training has begun focusing on decision-making procedures supported by data-driven methods (*Zhang et al., 2015*; *Mao, Zhang & Feng, 2008*). The training data of actual flights is typical multivariate time series (MTS) data (*Zhang et al., 2016*).

Assume that N different parameters represent the full aspect of flight actions. Some of them are called heading angles and pitch angles. Suppose the scores of M observations are recorded in order with a specific frequency. In that case, it is represented by n-ary time series data whose length is N, denoted by a design matrix, $A \in R$, whose order is MxN (*Yang & Xie, 2005*). Time series data can be described as a combination of non-stationary and stationary features (*Yabin, 2015*). The generic form of time series analysis is called multivariate time series analysis, a substantial method for crunching time-dependent datasets.

On the other hand, neural networks (NNs), an essential branch of AI, have also been implemented to identify flight actions. NNs have been implemented to forecast several flight actions. Researchers implement NNs to derive more determined insights based on generic aviation research.

# NEURAL NETWORKS

## The selection process of suitable networks

Deep learning (DL) algorithms represent the forefront of AI architectures, enabling knowledge acquisition from datasets. They are derived from application domains such as autonomous driving, robotics, facial recognition, natural language understanding, and other areas. However, contemporary AI models rely on extensive datasets for comprehension and struggle to transfer their domain-specific knowledge to new contexts. The essential DL algorithms contain convolutional neural networks (CNNs), recurrent neural networks (RNNs), long short-term memory networks (LSTMs), and generative adversarial networks (GANs). For example, CNN excels in image recognition by emulating the human brain's visual organization through a sophisticated DL architecture. RNN operates across multiple time steps and shares weights over time. However, the gradient-vanishing issue hinders its effective learning capability from longer sequences during backpropagation. Alternatively, LSTM presents a better solution to address the issue of short-term memory limitations by introducing internal mechanisms called gates that regulate the flow of information within long chains to facilitate accurate predictions.

The research employs a bi-directional RNN algorithm tailored to the characteristics of research objects to recognize flight actions. The bi-LSTM, an extension of RNNs and LSTMs, excels at grabbing long-term dependencies when dealing with sequenced data. Given that aircraft flight action data is characterized as a typical time series dataset, the selection of the bi-LSTM becomes imperative due to its capability to classify actions based on multiple parameters determined by their distinct characteristics across different periods.

The bi-LSTM crunches data sequentially, effectively capturing contextual information within the sequence and exhibiting enhanced generalization capabilities. Compared to the LSTM algorithm, the bi-LSTM algorithm incorporates reverse temporal details, allowing it to encode both past and future context simultaneously. It enables a more comprehensive grasp of the sequence's feature dynamics and facilitates a more precise prediction of its subsequent states.

## The bi-directional long and short-term neural networks

The bi-LSTM algorithm comprises input, forward backward, and output layers, respectively as shown in Fig. 1.

The LSTM algorithm comprises three control gates: forget, input, and output, respectively. The forget gate regulates the retention or deletion of information in the cell state, which computes a score based on the previous hidden state and current input at each step to find which information needs to be forgotten or preserved. The input gate determines the relevance of incoming information to update the current cell state, while the output gate controls the generation of the next hidden layer state (*Yang & Xie, 2005*; *Chuan et al., 2004*).

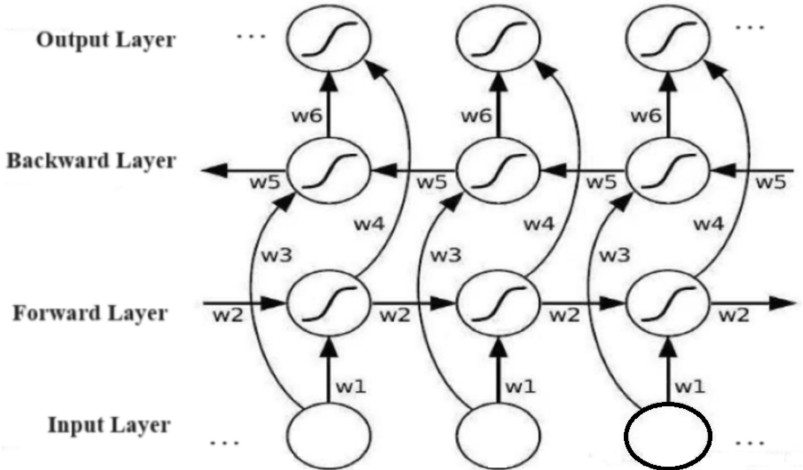

The bi-LSTM algorithm, utilizing the interplay among neurons, forget gates, memory gates, output gates, and hidden states, performs better in processing sequential data as it captures both long and short-term dependence within the sequence (*Jia et al., 2018*).

## Deep learning

RNN generally holds internal information. Prior input information, a substantial dataset, is utilized to estimate the subsequent information precisely. A network called Memory is suggested, which functions as an LSTM containing gates, namely, forget, input and output, and memory state structure, respectively. Elements' bitwise operations are represented by red, the network layer by yellow, and the cell state by the arrow, where the information is stable. The score that will be updated is determined by the input gate layer, and then a new candidate score vector is constructed by a tanh layer, which is supplemented with the state. Subsequently, the cell state is streamlined, and eventually, the output gate determines the output by employing the cell state. Equation (1) presents it.

$$f_i = \sigma(w_f[x_i, h_{i-1}] + b_f). \tag{1}$$

Equation (2) presents the forgetting gate's equation and has $x_i$ and $h_{i-1}$ outputs. The score of each number in the cell state $C_{i-1}$ has either 1, "fully kept", or 0, "fully discarded".

$$I_i = \sigma(w_I[x_i, h_{i-1}] + b_I) \tag{2}$$
$$C = \tanh(W_C[x_i, h_{i-1}] + b_c). \tag{3}$$

In Eqs. (3) and (4), C denotes the new scores of the vector and $I_i$ finds what score will be streamlined.

$$C_i = f_i^* C_{i-1} + I_i^* C_i. \tag{4}$$

Equation (5) designates that the emotional state of an individual alters from $C_{i-1}$ to $C_i, f_i^* C_{i-1}$, Indicating that the new vector must be discarded.

$$o_i = \sigma(W_0[x_i, h_{i-1}] + b_0) \tag{5}$$
$$h_i = o_i^* \tanh(C_i). \tag{6}$$

Equation (6) denotes the state to be outputted and then is processed by Eq. (8) to attain a score in $[-1, 1]$ and find the output.

## LSTM with attention mechanism

NNs in the standard memory form can no longer extract emotions from sentences. The attention mechanism applied in DL functions resembles human beings' selective attention mechanisms. As distinct emotions loom, the sentences' significant sections are grasped. The objective of the attention mechanism is to pick up more substantial information but ignore irrelevant ones when dealing with a problem. Thus, the computational burden of DLs can also be reduced. A modified LSTM with an attention mechanism that grasps pivotal sections of sentences is suggested to resolve the issue. Equations (7) through (10) are presented as follows:

$$M = \tanh\left(\begin{bmatrix} W_h H \\ W_v v_a \otimes e_N \end{bmatrix}\right) \tag{7}$$
$$\alpha = softmax(w^T M) \tag{8}$$
$$r = H\alpha^T \tag{9}$$
$$h^* = \tanh(W_{n'} + W_x h_N) \tag{10}$$

where $M \in R^{(d+d_\alpha)*N}$ and $N$ show the sentence's sequence, H denotes the input sentence's hidden note, Eq. (12) is employed to predict the result, $W_{p'}$ and $W_x$ shows the learned parameters of the model, $h^*$ represent the attribute representation of the given sentence as the input, and $h_N$ characterizes the hidden vector of the last layer of the hidden layer.

## Deep neuro factorization machine

The number of dimensions is reduced by the factorization machine (FM). A linear model can be employed if low-order data is available, while NNs are nonlinear models with high-order data. However, estimating the parameters requires complex operations when the sparsity of the data is observed. Thus, a deep neuro factorization machine (DeepFM) is suggested to combine the DNN with the FM, allowing both low- and high-order features to interact. The DeepFM can execute training with an end-to-end structure, although no feature engineering is applied. Equation (11) presents the training stage.

$$y = sigmoid(y_{FM} + y_{DNN}) \tag{11}$$

where $y_{FM}$ shows the FM component's output, $y_{DNN}$ denotes the deep component's output, and $y \in (0, 1)$ represents the DeepFM's predictions. Va is expressed in Eq. (12).

$$y_{FM} = w, x + \sum_{i=1}^{d} \sum_{j=i+1}^{d} \langle V_i, V_j \rangle x_i . x_j \tag{12}$$

where the inner product $\langle w, x \rangle$ denotes the first-order feature and second-order cross-feature, respectively, $y_{DNN}$ is expressed in Eq. (14),

$$\alpha^{(l+1)} = \sigma\left(w^{(l)}a^{(l)} + b^{(l)}\right) \tag{13}$$

$$y_{DNN} = W^{|H|+1}.\alpha^{|H|} + b^{|H|+1} \tag{14}$$

where the activation function is represented by σ, the DNN's layer number is denoted by 1, $w^{(l)}$ shows the DFM's weight, $a^{(l)}$ denotes the output of layer l, $b^{(l)}$ represents the bias term, and $|H|$ denotes the hidden layer's numbers. A prediction is attained when Eq. (14) is used. An activation function computes neurons, representing the relationship between neurons' input and output.

The data in deep neurofactors is employed as input from deep to hidden layers. If errors occur, an error range is utilized to tune the parameters. To attain an output, distinct layers are continuously adjusted. Thus, the training stage is completed when parameters converge and show no change according to the previous criteria, and the average loss function is minimized. Equation (15) represents the loss function.

$$L_{reg} = \sum_{x \in D} \left(r_{u,j}(x) - r_{u,j}(x)\right)^2 \tag{15}$$

where D represents the training set, and $r_{u,i}(x)$ denotes the user's satisfaction score with series I.

## The input vector representation of the decomposer

The sum of squared errors, RMSE, is computed in Eq. (16) and MAE is represented by Eq. (17).

$$RMSE(x, h) = \sqrt{\frac{1}{m}\sum_{i=1}^{m}(h\left(x^{(i)} - y^{(i)}\right)^2}. \tag{16}$$

MAE is calculated by

$$MAE(x, h) = \frac{1}{m}\sum_{i=1}^{m}\left|h\left(x^{(i)} - y^{(i)}\right)\right|. \tag{17}$$

Both small RMSE and MAE scores designate more accurate results. Both accuracy and recall metrics are utilized to measure models' performance. Equations (18) and (19) present them.

$$Precision = \frac{TP}{TP + FP} \tag{18}$$

$$Recall = \frac{TP}{TP + FN} \tag{19}$$

where precision is calculated as a ratio of true positives over the sum of true positives and false positives in the prediction stage. Thus, the larger the accuracy, the better the performance and the better the algorithm's impact. Recall denotes the ratio of true

positives over the sum of true positives and false negatives in the prediction stage. Hence, the bigger the rate, the larger the coverage ratio, and the better the impact.

Prediction accuracy is the most broadly utilized assessment indicator presented by Eqs. (20) and (21), in the literature.

$$MAE = \frac{\sum_{n=1}^{N} |p_i - q_i|}{N} \tag{20}$$

$$RMSE = \sqrt{\frac{\sum_{n=1}^{N} (p_i - q_i)^2}{N}} \tag{21}$$

where {pl, p2... pn} represents the set showing the predicted recommendations, and the set {q1, q2... qn} shows the ratings of the real users. The smaller the RMSE and MAE, the smaller the error between the predicted and real scores, and the higher the recommendation quality. The lower quality of recommendation is attained contradictorily.

# IMPLEMENTATION

## The construction of the dataset

Flight data is typically time series data that records flight actions chronologically. The entire process of flight training, from takeoff to the shutdown of engines, includes several operations such as engine monitoring, trajectory (flight path), attitude information (flight attitude), air conditions (air-related data), and so on. The original flight training data is provided in a CSV format with timestamps and includes headers and corresponding data items. After running statistical analyses on the dataset, such as collinearity checks, correlation analyses, and cross-correlations, 21 usable and valid parameters are detected for each flight. The present research utilizes 105 CSV files, each with an approximate flight duration of 1.5 h.

## Data labeling

A system recognizing flight actions is utilized based on expert rules to forecast 105 sorties, 103 jacks, 141 half-jacks, and 153 half-roll reversals from the datasets.

## The formation of the dataset

The formation of the dataset includes four CSV files, namely, jd.csv, bjd.csv, bgsz.csv, and other.csv, that correspond to all jack data, all half jack data, all half roll reversal data, and the data of actions other than these three types. To optimize the reading speed, jd.csv, bjd.csv, bdz.csv, and other.csv are merged into a single file named merged_data.pkl.

## Parameter selection

1) The optimizer selection is based on the parameters that are not always convex functions but rather complex functions. The gradient descent algorithm may encounter local minima issues. Therefore, various gradient descent algorithms have been employed to deal with the issue, including stochastic gradient descent (SGD), mini-batch stochastic gradient descent (MBGD), momentum SGD, weighted average second-order Momentum (RMSProp), and the Adam optimizer, which combines multiple

**Table 1 Predictions (20030501).**

| Action types | Action 1 | Action 2 | Action 3 |
|---|---|---|---|
| Rule judgment | 2 | 3 | 4 |
| Intelligent recognition | 1 | 2 | 2 |
| Results | 1 | 1 | 2 |

optimization techniques. Essentially, the Adam optimization algorithm integrates both the advantages of the SGD and RMSProp while allowing different parameters to adapt to varying learning rates.

2) The activation function is also crucial for enriching the hypothesis space, showcasing the advantages of multi-layer representation, and introducing nonlinearity to each neuron, enabling the model to flexibly approximate any nonlinear function and form diverse nonlinear models with enhanced fitting capabilities. Commonly implemented activation functions include sigmoid, Tanh, ReLU, and Softmax. Finally, the ReLU activation function is picked in the research.

3) The selection of the loss function plays a crucial role in optimizing the constructed model and is an integral parameter. The appropriate choice of loss function depends on the problem characteristics and can be guided as follows: For regression-based models involving continuous-valued vectors, MSE is suggested to be employed. In binary classification problems, it is advisable to employ the binary cross-entropy loss function. For multi-class classification problems, categorical cross-entropy should be employed by utilizing one-hot encoding to represent the output. Whereas if integer values are employed for output representation, sparse categorical cross-entropy mapping should be implemented instead. Since the research involves multi-classification with integer outputs, sparse categorical cross-entropy mapping should be chosen to assess performance.

4) The selection of evaluation metrics or indicators is implemented to gauge the performance of the constructed model, functioning as loss functions that aim to minimize the error between true and predicted scores. However, their application differs. While loss functions guide gradient descent during training, evaluation metrics are utilized on validation and test sets to assess the constructed model's accuracy. They serve as the default evaluation metric for classification problems, representing the correctly classified proportion of samples. Therefore, the accuracy rate is adopted as the evaluation index.

## Analysis of the training stage and results

A GEFORCE RTX4080 GPU computer is utilized to conduct the research. The training set is implemented to construct the model, and parameters are iteratively adjusted through repeated samples; an initial model is eventually attained. To validate the accuracy of the

constructed algorithm, the whole flight action dataset is implemented by utilizing the constructed model to generate classification results. Table 1 presents the outcomes.

Table 1 suggests that the intelligent detection effect does not satisfy the indexing requirement. By Rule Judgement, there should be 2 Action 1s. However, the thoughtful recognition has not yet been judged. Action 2 decides by Rule Judgement that there should be three moves; however, it identifies 2. Action 3 utilizes the Rule Judgement that there should be four actions, and the intelligent recognition finds only 2, which is correct. The accuracy of current intelligence recognition accounts for just 30 percent of the data.

Action recognition is faulty since there could be possible reasons, namely, the bi-LSTM does not master the original data sufficiently. The allocated training set consists of a set of individual actions. The model's compatibility is not excellent if full data is employed as the input. The model does not segment the data adequately, resulting in worse action recognition. Also, the amount of data is limited. Therefore, the algorithm needs further improvement.

# MODEL IMPROVEMENT AND OPTIMIZATION

The algorithm must be optimized to enhance accuracy and simultaneously reduce learning time.

## The enhancement of the dataset

The training set is augmented to make the algorithm more sensitive to actions. There are several methods to enhance the capability of signals. Commonly and efficiently implemented data augmentation methods include time translation, data rotation, time scaling, and data truncation, which could effectively enhance the diversity and robustness of the data set and the performance of constructed models when the data is characterized as a time series. So, the Addnoise() and pool () methods are employed to add noise to the dataset to enrich time series data if needed.

The Addnoise () method is implemented to supplement noise to time series data to enhance the robustness and generalization capability of the data. It can help the constructed model master noise and uncertainty in real-world actions.

The code is presented as follows:

```
def test_tsaug():
    import tsaug
    from tsaug.visualization import plot
    X = np.array([[[1,2],[2,3],[3,4]]])
    Y = np.array([[1,2,3]])
    X_aug, Y_aug = tsaug.AddNoise(scale=0.01).augment(X, Y)
    X_aug, Y_aug = tsaug.Quantize(n_levels=10).augment(X, Y)
    X_aug, Y_aug = tsaug.Pool(size=2).augment(X, Y)
    plot(X_aug, Y_aug);
```

Also, pooling is commonly employed to reduce data dimensions, extract crucial features, and improve data efficacy. It involves scanning a window of matrices across the tensor and reducing the number of elements in each matrix by selecting either the maximum or average score. This technique enables feature extraction to possess "translation invariance," meaning that a stable combination of features can still be obtained even with slight pixel displacement in an image. In models, the pooling function typically resides one layer below the convolution function and diminishes dimensionality. Popular pooling functions include Average and Max pooling. Max pooling selects the most significant score within a window as output, while the average is computed by utilizing all scores within a window as output. These operations effectively decrease the number of model parameters while retaining essential features, thereby simplifying the structure of the constructed model.

The tsaug augmentation library is implemented in Python 3.2. In time series data augmentation, Pool (size = 2) refers to the pooling operation where every two data points of the time series data are merged into one data point. Specifically, it merges every two data points adjacent to it into a different data point for each, thus reducing the number of data points. This operation can effectively lower the size of the dataset, and the pooling operation can also be employed to downsample the dataset to manage large-scale datasets better. AddNoise (scale = 0.01) is the operation that supplements noise to the data. Noisy data is generated by utilizing a specified noise ratio (scale = 0.01), then supplemented with the original data to produce the augmented data. By adding noise, possible perturbations and uncertainties in the actual data could be better simulated, thus increasing the diversity and generalization power of the constructed algorithm.

## The data splitting process

In the previous training set, the training set of segmented actions was sent to the model for learning, which can improve learning efficiency. Still, the generalization capability of the algorithm was not strong. Thus, if the data is segmented in advance and sent to the constructed model for mastering, it can lower the inference time and enhance the model's classification precision.

The article attempts to segment the data by employing the sliding window method. Generally, methods used for splitting time series data are based on sliding windows that mainly split the data set into multiple substances. So, a fixed-length window is set and is slid over the data set. The steps are presented as follows:

(1) Determine the sliding window's width and step size. The sliding window's width can be assigned according to the requirements and characteristics of the dataset, and the step size can be set according to the frequency and sampling interval of the data.

(2) Flatten the data set. The data set is flattened chronologically to form a one-dimensional data sequence.

(3) Use a sliding window to segment sequence data. A sliding window is moved over the sequence data, split into multiple subsequences.

Each subsequence is then processed, which can be fed into the model for training, prediction, or processing. Note that when splitting time series data by implementing a sliding window, the following points should be noted:

The width and step size of the sliding window need to be set according to the characteristics of the data set to obtain better segmentation results. The segmented subsequences should be of sufficient length and stability for further processing and analysis.

The application of sliding windows can be combined with other data processing methods, such as feature extraction and normalization, thus further improving segmentation effects and processing efficiency. Data segmentation is performed based on the regular aspects of the data features.

To avoid missing the actions of an aircraft, the goal is to detect it whenever its state changes. The subject implements qualitative and quantitative analysis methods by plotting line graphs, extracting data extremes, and computing variances, maxima, and minima. The action can also be decomposed into combinations of different flight parameters when further analysis is applied to the data features. This topic focuses on flight parameters that have a large impact on the identification of flight actions, including two features: pitch angle and tilt angle. Through repeated observations and trials, the boundary conditions of the sliding window are set based on these two characteristics.

(1) Determine the number of the ten consecutive absolute values of the pitch angle greater than 20 is greater than or equal to 2. If the pitch angle is greater than 10, it is cut. Also, the cut is terminated when six consecutive absolute values of the tilt angle less than 3.5 are greater than or equal to 4, and six consecutive absolute scores of the pitch angle less than 6 are greater than or equal to 4.

```
        Algorithm:
    flag3_begin = np.sum(np.abs(df['pitch angle'].iloc[index:index + 10].values) > 20) >= 2
and fuyang > 10
     flag3_end = np.sum(np.abs(df['tilt angle'].iloc[index:index + 6].values) < 3.5) >= 4 and
np.sum(
                np.abs(df['pitch angle'].iloc[index:index + 6].values) < 6) >= 4
```

(2) If the tilt angle's absolute score is greater than 2 or 4 for five consecutive values, the cut is terminated. When six consecutive absolute values of the tilt angle are less than 3.5 and are greater than or equal to 4, and six consecutive absolute values of the pitch angle are less than 6 and are greater than or equal to 4, the cut is terminated.

A data segmentation method is proposed based on a sliding window. The program is implemented as follows:

```
def roll_append(df,action,flag,index,ignore_value):
if flag == 'begin':
        window_data = df.iloc[index − ignore_value:index].values
        action.extend(window_data)
```

**Table 2 Loss and accuracy scores.**

| Input_sizes | Train_loss | Val_loss | Train_acc | Val_acc |
| --- | --- | --- | --- | --- |
| 11 | 0.0703 | 0.380 | 0.897 | 0.852 |
| 5 | 0.535 | 0.410 | 0.901 | 0.900 |
| **3** | **0.0053** | **0.208** | **0.980** | **0.903** |

Note:
   The bold numbers are the best values attained when Input size, Drop_out, and epoch change.

**Table 3 Dropout scores and loss scores.**

| Drop_out | Train_loss | Val_loss | Train_acc | Val_acc |
| --- | --- | --- | --- | --- |
| 0.5 | 0.517 | 0.375 | 0.972 | 0.894 |
| 0.3 | 0.0524 | 0.399 | 0.975 | 0.898 |
| **0.1** | **0.0053** | **0.208** | **0.980** | **0.903** |

Note:
   The bold numbers are the best values attained when Input size, Drop_out, and epoch change.

**Table 4 Presents results when different epochs are utilized.**

| Epoch | Train_loss | Val_loss | Train_acc | Val_acc |
| --- | --- | --- | --- | --- |
| 30 | 0.114 | 0.398 | 0.960 | 0.872 |
| 50 | 0.071 | 0.372 | 0.978 | 0.885 |
| **100** | **0.0053** | **0.208** | **0.980** | **0.903** |

Note:
   The bold numbers are the best values attained when Input size, Drop_out, and epoch change.

**Table 5 Batch_sizes and loss scores.**

| Batch_size | Train_loss | Val_loss | Train_acc | Val_acc |
| --- | --- | --- | --- | --- |
| 32 | 0.0387 | 0.350 | 0.977 | 0.890 |
| **64** | **0.0053** | **0.208** | **0.980** | **0.903** |
| 128 | 0.0447 | 0.377 | 0.978 | 0.901 |

Note:
   The bold numbers are the best values attained when Input size, Drop_out, and epoch change.

**Table 6 Comparison of different methods.**

| File names | Methods | Action 1 | Action 2 | Action 3 |
| --- | --- | --- | --- | --- |
| 231010002 | Rules | 1 | 0 | 0 |
|  | BILSTM | 1 | 1 | 0 |
| 231010003 | Rules | 0 | 0 | 2 |
|  | BILSTM | 0 | 0 | 6 |
| 231010004 | Rules | 0 | 0 | 0 |
|  | BILSTM | 0 | 0 | 0 |
| 231010005 | Rules | 1 | 0 | 0 |
|  | BILSTM | 2 | 1 | 1 |
| 231010006 | Rules | 2 | 2 | 1 |
|  | BILSTM | 4 | 2 | 2 |

```
if flag == 'end':
    window_data = df.iloc[index + 1:index + ignore_value].values
    action.extend(window_data)
```

### Normalisation

In general, normalization transforms all data into a specific range, such as [0, 1] or [−1,1].

### Numerical results

Previously, the internal parameters of the model were only tuned. After the previous data processing, we continued to adjust the parameters of the bi-LSTM.

The input_size denotes the input dimension, namely, how many attributes exist in each row. Parameter numbers 11, 5, and 3 are implemented to calculate the loss and accuracy rates for the training and test sets, respectively. The results are presented in Table 1. The internal parameters of the model are only adjusted. Then, the bi-LSTM-related parameters are tuned after the previously conducted data processing, as shown in Table 2.

Table 3 presents drop_outs that refer to randomly dropping some neurons during the training stage to prevent overfitting. Table 4 shows the results of utilizing different epochs.

Table 5 summarizes the samples used in each training iteration of the layer.

## CONCLUSION

After the constructed model is optimized, the training and test sets have loss rates of 0.0053 and 0.208, and accuracy of 0.98 and 0.903 are obtained for the training and test sets, respectively. To ensure the generalizability of the constructed model, five additional flight data files for the parameters are selected, and the results that employ the rule and the BILSTM model are presented in Table 6.

In conclusion, the bi-LSTM cannot only forecast all the actions identified by the rule AS but also irregular movements. Meanwhile, the segmentation approach can enhance recognition precision and distinct segment actions.

### Funding

The authors received no funding for this work.

### Competing Interests

The authors declare that they have no competing interests.

### Author Contributions

- Xin Hua conceived and designed the experiments, performed the experiments, analyzed the data, performed the computation work, prepared figures and/or tables, authored or reviewed drafts of the article, and approved the final draft.

- Xuejie Yang conceived and designed the experiments, performed the experiments, analyzed the data, performed the computation work, prepared figures and/or tables, authored or reviewed drafts of the article, and approved the final draft.

## Data Availability

The data and code is available in the Supplemental Files.

## Supplemental Information

Supplemental information for this article can be found online at http://dx.doi.org/10.7717/peerj-cs.2153#supplemental-information.

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
