# Peer review of "Research on intelligent forecasts of flight actions based on the implemented bi-LSTM"

_PeerJ Computer Science, doi:10.7717/peerj-cs.2153_

## Round 0.1 · original submission · Major Revisions

Dear authors

Thanks for your submission, based on the input received from the reviewers, I'm afraid that we cannot consider your article in its current form, you can see that some major issues needed to be addressed as mentioned below. Therefore , we invite you to carefully revise the paper and submit a detailed response letter and the changes impact in the paper. Thanks

Reviewer 1 ·

Basic reporting

The paper proposes a method for rapid identification of aircraft flight actions using the Bidirectional Long Short-Term Memory (bi-LSTM) algorithm. The aim is to objectively assess the quality of flight training by accurately recognizing flight actions based on real flight data, overall, the paper seems to find that the incorporation of the following suggestions could make it more strong.

1. Please Provide more details on how the dataset of flight training actions is constructed, including the specific flight data collected, the criteria for accurate data selection, and the expert rules used for annotation

2. Expand on the preprocessing techniques applied to the dataset before training the bi-LSTM algorithm

3. While it's mentioned that pivotal parameters of the bi-LSTM algorithm are optimized, provide more insight into the optimization process

4. Title: The word recognition is used to identify patterns in the data, however, forecasting is more technical to support future predictions in time series data. Bi-LSTM is a model that could be represented in mathematical form so, the word model is more appropriate than the algorithm while using Bi-LSTM.

6. Bidirectional Recurrent Neural Networks were proposed in 1997 by M.Schuster and gained prominence in 2010 for sequence modeling tasks. The authors discussed and claimed their model named “Bi-LSTM Algorithm” which is controversial. They need to highlight their contributions to the advancement of Bi-LSTM for flight action forecasting.

7. Bi-LSTM doubles the computation cost as compared to simple LSTM and you have claimed about short computation time for large samples. The authors focus on describing Bi-LSTM and do not highlight how their proposed model minimizes computation cost.

8. The deep learning models require many data instances for training and testing, the authors mentioned 105 data files from which each file has 1.5 hours of flight data. The total data instances or time interval to collect this data in these files is not mentioned.

Experimental design

The detailed comments have been reported in the basic reporting section.

Validity of the findings

The detailed comments have been reported in the basic reporting section.

·

Basic reporting

The overall presentation of the paper is below the standards expected for publication in PeerJ. The manuscript contains numerous typographical errors and lacks a clear motivation for the research conducted.

In particular, the explanation provided in Section 3.3 is oversimplified and does not effectively connect with the problem domain. The transition from Equation 1 to Equation 6 is unclear and difficult to follow. Sections 3.3 to 3.5 suffer from readability issues, making it challenging even for readers with a strong background in the field to comprehend the content.

Equation 16 is particularly ambiguous. It would be beneficial to represent "popularity" using a mathematical symbol rather than a complete word. Additionally, the interpretation of AS_i as "A x S_i" is incorrect and misleading. It is imperative that all equations are revised and rewritten for clarity and accuracy.

Experimental design

The primary contribution of this work appears to be the dataset used in conjunction with BiLSTM. However, employing BiLSTM on a challenging dataset might not be suitable given the age of the model.

Could you provide more details on how the dataset was acquired?

Sections 4.1 to 4.4 require comprehensive rewriting for clarity. Despite multiple readings, I struggled to fully grasp the content.

What statistical analyses were conducted to identify the 21 usable parameters for each flight? In Section 4.2, the actions referenced are not adequately explained in the text.

Could you clarify the size of the dataset, including the number of instances and class distribution? Providing these details is essential to establishing the impartiality and authenticity of the dataset.

Table 1 appears too simplistic and lacks clear connection to the accompanying explanation.

The absence of a baseline model for comparison is notable. Given the basic implementation and high validation accuracy, the credibility of the problem may be undermined. I recommend experimenting with other models on the same dataset to gauge their effectiveness. If all models achieve accuracies above 85%, it suggests that the dataset may be too simplistic.

Validity of the findings

Illustration of findings is too naive, as I mentioned above, there is no baseline. You used Rule based but didn't bother to explain that. Table 6 is references in conclusion ?

---

## Round 0.2 · Minor Revisions

Dear authors, your revised version has been reviewed by the experts and based on their input, I am pleased to inform you that your manuscript has been scientifically okay , but few observations need your attention before we can further consider it.

1. Please carefully check and improve the language of the article so that it should be easy and understandable to the readers. Eg making sentences clear and unambiguous, making long sentences short and correct the spelling and grammar mistakes.

2. Please cross check the data and code files and make sure that they are correct, openable and easy to understand for reproduction.

Thanks

Reviewer 1 ·

Basic reporting

I am satisfied with the revisions and now the paper may be considered for acceptance. However, it is recommended that the authors proofread the text using a professional language editor to improve the paper text further.

Experimental design

No further comments.

Validity of the findings

No further comments.

Additional comments

No further comments.

·

Basic reporting

All comments have been addressed.

Experimental design

All comments have been addressed.

Validity of the findings

All comments have been addressed. However, the source code and the dataset provided can't be opened--at least on my computer.

---

## Round 0.3 · accepted · Accept

Thankyou for addressing the comments of second round. In pleased to inform you about the acceptance of your article.